# HiRes-GS: Hierarchical Resolution Scaling for Sparse-View High-Resolution 3D Gaussian Splatting

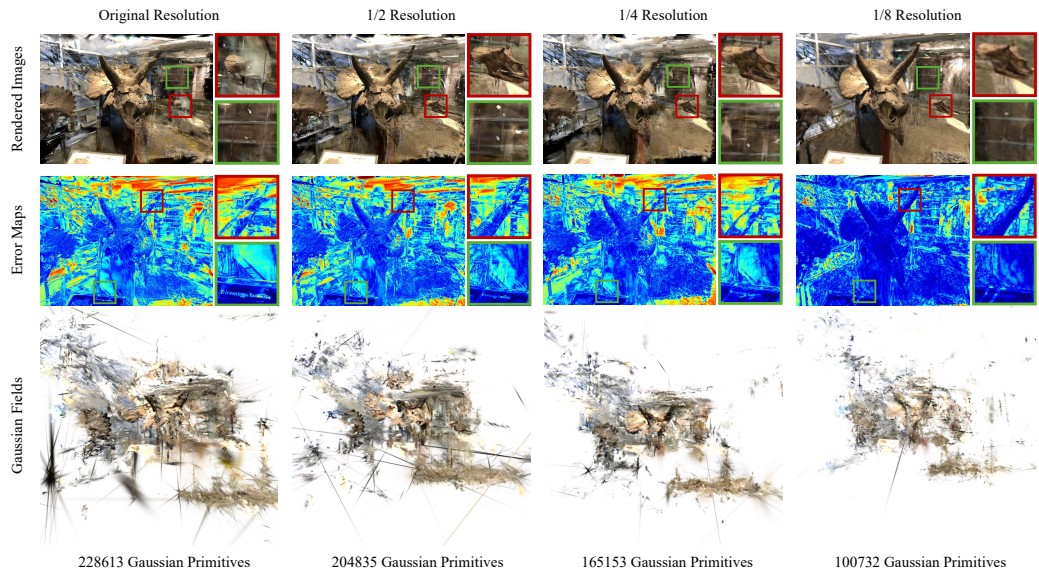

Figure 1: **Empirical study under different input resolutions.** The rendered images, error maps, and Gaussian fields reveal a multi-resolution complementarity: lower resolutions produce robust global structures but lose fine details, while higher resolutions capture finer features yet introduce more noise and ghosting.

## ABSTRACT

Sparse-view 3D Gaussian Splatting (3DGS) reconstructs scenes using 3D Gaussians from sparse input views. Yet, this method is prone to overfitting, which is exacerbated at higher resolutions as the expanded dimensionality amplifies floating artifacts and reconstruction ambiguities. In this paper, we systematically investigate the reconstruction performance of 3DGS in sparse views under varying resolutions, and **we are the first to discover that low-resolution inputs yield stable global structures but lose fine details, while high-resolution inputs capture richer local features at the cost of increased noise and ghosting.** Motivated by this finding, we further propose **HiRes-GS**, a hierarchical multi-scale paradigm that reconstructs scenes through a coarse-to-fine hierarchical optimization process. Our approach employs a matching-based pruning strategy to anchor high-resolution reconstructions to stabilize structural priors and filtering noise through cross-scale consistency, and a multi-scale pseudo-view regularization to refine local details without amplifying noise. Extensive experiments on the LLFF and Mip-NeRF360 datasets demonstrate that HiRes-GS significantly outperforms existing methods, particularly under demanding high-resolution conditions. Moreover, our paradigm can be seamlessly integrated into other 3DGS-based pipelines, thereby extending the field from low-resolution reconstructions to high-fidelity outputs under real-world sparse-view constraints.

# 1 INTRODUCTION

In recent years, significant progress has been made in 3D scene reconstruction, with methods such as Neural Radiance Fields (NeRF) Mildenhall et al. (2021) and 3D Gaussian Splatting (3DGS) Kerbl et al. (2023) enabling photorealistic novel view synthesis for a variety of applications, such as autonomous driving Yu et al. (2024); Zhou et al. (2024) and 3D content creation Tang et al. (2023; 2024); Liu et al. (2024). As these technologies mature, the demand for high-resolution reconstructions has grown, driven by the need for detailed and visually compelling outputs in real-world scenarios. However, when limited input views are available, traditional methods tend to overfit training views, leading to degraded performance and unstable geometry. Recent works Zhu et al. (2024); Chung et al. (2024); Zhang et al. (2024); Deng et al. (2022); Kim et al. (2022); Niemeyer et al. (2022); Yang et al. (2023) have sought to alleviate the challenges of sparse-view reconstruction using downsampled input. Despite these advances, achieving high-resolution outputs remains a formidable challenge, as the optimization of high-resolution images, with their abundant fine-scale details, is inherently more difficult.

A major challenge in sparse-view scenarios lies in the densification process of 3D Gaussian Splatting, which transforms a sparse initial point cloud into a more detailed scene representation. However, with limited input views, the placement of new Gaussians tends to produce noisy and unrepresentative. Although recent approaches such as FSGS Zhu et al. (2024) and CoR-GS Zhang et al. (2024) have made notable progress in addressing these issues by incorporating additional priors and co-regularization strategies on downsampled datasets (e.g., $8\times$ reduction), their effectiveness diminishes in high-resolution reconstruction. At higher resolutions, the addition of more details not only leads to a proliferation of Gaussian primitives but also amplifies overfitting and ghosting artifacts, thereby degrading the overall reconstruction quality. This persistent challenge underscores the need for novel approaches that can effectively balance global structural stability and local detail refinement under sparse-view conditions.

We delve into these challenges, conducting a systematic investigation of how varying input resolutions affect reconstruction quality under sparse-view conditions (see section 3.1 for details). Our experiments reveal that low-resolution inputs, while sacrificing certain high-frequency details, yield more stable global structures and suffer less from overfitting. In contrast, higher-resolution inputs provide richer local details but become increasingly prone to noise and ghosting artifacts, particularly in under-constrained regions. These observations suggest a natural multi-resolution complementarity, wherein lower resolutions act as a robust anchor for geometry, and higher resolutions enhance fine-scale features. Motivated by this insight, we propose a *Hierarchical Multi-scale Paradigm* that integrates coarse and fine Gaussian fields, progressively leveraging the strengths of each resolution level to achieve high-resolution reconstructions, even in sparse-view scenarios.

In this work, we propose **HiRes-GS**: a 3DGS-based method that fuses the robust global structure provided by low-resolution inputs with the fine-grained local details captured at high resolution, thereby enabling high-resolution scene reconstruction under sparse-view conditions. Specifically, our method constructs multiple 3D Gaussian fields at different resolutions and employs a hierarchical paradigm in which the coarser (low-resolution) fields serve as stable anchors to progressively guide the refinement (pruning) of the finer (high-resolution) fields.To further mitigate noise and overfitting, we apply a multi-scale pseudo-view regularization strategy. Our experimental results on the LLFF and Mip-NeRF360 datasets demonstrate significant improvements in high-resolution scene reconstruction, effectively bridging the gap in sparse-view reconstruction with high-resolution inputs. Furthermore, our paradigm can be seamlessly integrated into existing 3DGS pipelines, empowering them with a coarse-to-fine hierarchical optimization mechanism. The paradigm design ensures compatibility with diverse 3DGS implementations, enabling systematic refinement from global shape recovery to fine-grained detail synthesis.

The main contributions of this paper are summarized as follows.

- We are the first to analyze the impact of input resolution on sparse-view reconstruction, revealing that low-resolution inputs yield robust global structure, while high-resolution inputs capture finer details but are more prone to noise and overfitting.

- We propose a hierarchical paradigm that progressively prunes multi-resolution Gaussian fields, integrating robust low-resolution structure with fine high-resolution details to bridge the gap in sparse-view high-resolution reconstruction.

- Our generic framework enhances existing 3DGS methods via coarse-to-fine hierarchical optimization, universally improving detail reconstruction across sparse-view settings. Our experiments demonstrate more complete structures, richer details, and fewer floaters across multiple benchmarks compared to state-of-the-art approaches.

## 2 RELATED WORKS

### 2.1 RADIANCE FIELDS FOR 3D RECONSTRUCTION

Radiance fields have been extensively used to reconstruct 3D scenes Barron et al. (2022); Müller et al. (2022); Martin-Brualla et al. (2021); Zhou et al. (2024) and generate novel views Chen et al. (2021); Chibane et al. (2021); Wang et al. (2022); Johari et al. (2022). Recent advances in neural rendering techniques, such as Neural Radiance Fields (NeRF) Mildenhall et al. (2021), represent a key breakthrough in this domain, learning a neural volumetric representation that maps 3D coordinates to radiance and density, subsequently rendered via volume integration. Since their introduction, extensive research has been dedicated to enhancing rendering fidelity Barron et al. (2021; 2022; 2023); Guo et al. (2022); Chen et al. (2022b) and efficiency Chen et al. (2022a); Fridovich-Keil et al. (2022); Hu et al. (2023); Liu et al. (2020); Müller et al. (2022); Sun et al. (2022); Yu et al. (2021). However, despite their impressive performance, these NeRF-like models often require several hours of training time. Recent advancements Kerbl et al. (2023); Chen et al. (2023); Xu et al. (2022) in unstructured radiance fields explored representations based on discrete primitives. Among these, 3D Gaussian Splatting (3DGS) Kerbl et al. (2023) stands out by representing scenes through anisotropic 3D Gaussians and performing image generation via differentiable splatting. Various recent 3D tasks Wu et al. (2024); Luiten et al. (2024); Tang et al. (2023); Zhou et al. (2024) that utilize 3DGS have achieved remarkable results in real-time reconstruction of complex tasks. Building upon this framework, our proposed HiRes-GS approach focuses on high-fidelity scene reconstruction while significantly reducing the number of training views required.

### 2.2 NOVEL VIEW SYNTHESIS WITH SPARSE VIEWS

While traditional 3D reconstruction approaches typically require more than one hundred images as input Avidan & Shashua (1997); Zhou et al. (2016), limiting their practicality, recent efforts have aimed to tackle the more challenging reconstruction of a few shot scenes Chung et al. (2024); Li et al. (2024); Paliwal et al. (2024); Zhu et al. (2024). Previous studies have investigated methods for regularizing NeRFs under sparse view conditions Deng et al. (2022); Kim et al. (2022); Niemeyer et al. (2022); Yang et al. (2023), paving the way for recent research.

With the advancements of 3DGS, FSGS Zhu et al. (2024) was among the first to integrate 3DGS for few-shot scene reconstruction from our knowledge, introducing a proximity-based Gaussian unpooling strategy that increases the overall density of the Gaussian distribution. DNGaussian Li et al. (2024) employs a pretrained monocular depth estimator Ranftl et al. (2021) to guide scene geometry refinement, while DepthRegGS Chung et al. (2024) combines depth predictions from monocular models Bhat et al. (2023) with sparse point cloud depths to regularize the radiance field. CoherentGS Paliwal et al. (2024) utilizes a learnable implicit decoder to enforce coherence across Gaussians. CoR-GS Zhang et al. (2024) proposed a co-regularization framework that trains two parallel 3DGS models, mitigating inconsistencies by suppressing unreliable regions through point and rendering disagreements.

Despite these advances addressing the few-shot reconstruction problem to some extent, they typically rely on significant downsampling (e.g., $8\times$ downsampling) to prevent floaters and inconsistencies at higher resolutions. Therefore, few-shot reconstruction in high-resolution settings remains a challenging problem.

# 3 EMPIRICAL STUDY

## 3.1 MULTI-RESOLUTION COMPLEMENTARITY

To investigate the effects of varying input resolutions under sparse-view conditions, we conduct a series of experiments using 3D Gaussian Splatting (3DGS) Kerbl et al. (2023) on images at different resolutions: the original high resolution, and downsampled versions at 1/2, 1/4, and 1/8. As shown in fig. 1, we compare the rendered images, corresponding error maps (where deeper blues indicate smaller errors and reds indicate larger errors), and the final Gaussian fields.

As observed in the rendered images (top row), reconstructions from lower-resolution inputs tends to maintain more coherent global structures with the smoother error maps (middle row) indicating more uniformly stable geometries. Nevertheless, they fail to preserve high-frequency details, resulting in noticeable blurring. Conversely, higher-resolution inputs exhibit substantially richer details but are significantly more susceptible to ghosting artifacts, disrupting overall structural integrity, as evidenced by fragmentary red regions in the error maps. This limitation arises from the sparse-view constraint, which impedes the accurate optimization of a dense set of high-resolution Gaussian primitives.

To better illustrate this trade-off, we visualized the Gaussian fields in the bottom row of fig. 1. Notably, higher resolutions introduce significant noise and redundant clusters, negatively affecting structural clarity. While coarser resolutions contain fewer points, they still provide a stable global structure. These observations underline an inherent complementarity between multi-resolution representations, therefore motivate us to explore a strategy that integrates the strengths of both ends of the spectrum. A straightforward approach is to use low-resolution fields to anchor stable geometry, selectively constraining the distribution of high-resolution Gaussian fields to preserve structural details while filtering out noise.

# 4 METHOD

Our goal is to exploit the complementary strengths of low- and high-resolution reconstructions to achieve high-fidelity 3D Gaussian radiance fields under high-resolution, sparse-view conditions. fig. 2 provides an overview of our pipeline, which proceeds in three stages. First, we build a *hierarchical multi-scale paradigm* (§4.1) by generating multiple Gaussian fields from downsampled versions of the input images; coarser scales supply robust global geometry, while finer scales capture richer local details. Second, we utilize a *matching point and pruning* mechanism (§4.2) that uses a coarser-scale field as an anchor to remove unreliable Gaussians at the finer scale, thereby mitigating overfitting and noise while preserving fine details. Finally, we impose a *multi-scale regularization* step (§4.3) by rendering pseudo-views across different resolutions and enforcing consistency with a high-resolution reference. This combination of hierarchical fusion, pruning, and regularization allows us to preserve both global stability and local detail, substantially improving reconstruction quality in sparse-view, high-resolution settings.

## 4.1 HIERARCHICAL MULTI-SCALE PARADIGM

As discussed in Empirical Study, low-resolution reconstructions provide robust global geometry while high-resolution inputs excel at capturing fine-grained details. Building on these complementary strengths, we propose a hierarchical multi-scale framework that fuses different resolution levels to improve the fidelity of sparse-view 3D reconstruction. Specifically, we generate multiple downsampled versions of the input images (e.g., 1/16, 1/8, 1/4, 1/2), each of which produces a corresponding 3D Gaussian radiance field. This process yields a sequence of Gaussian fields, from the coarsest (offering stable overall structure) to the finest (capturing high-frequency details).

As illustrated in the left and middle of fig. 2, we treat the lowest-resolution Gaussian field as a global anchor that informs higher-resolution refinements. By relying on the coarse-scale's stability, we mitigate the local inconsistencies often observed in high-resolution reconstructions. Consequently, we start with the coarsest Gaussian field, ensuring it converges to a reliable global representation, and then progressively incorporate the next finer-scale fields in a bottom-up manner. At each stage, the stable geometry from the coarser resolution serves as a reference, allowing us to preserve beneficial

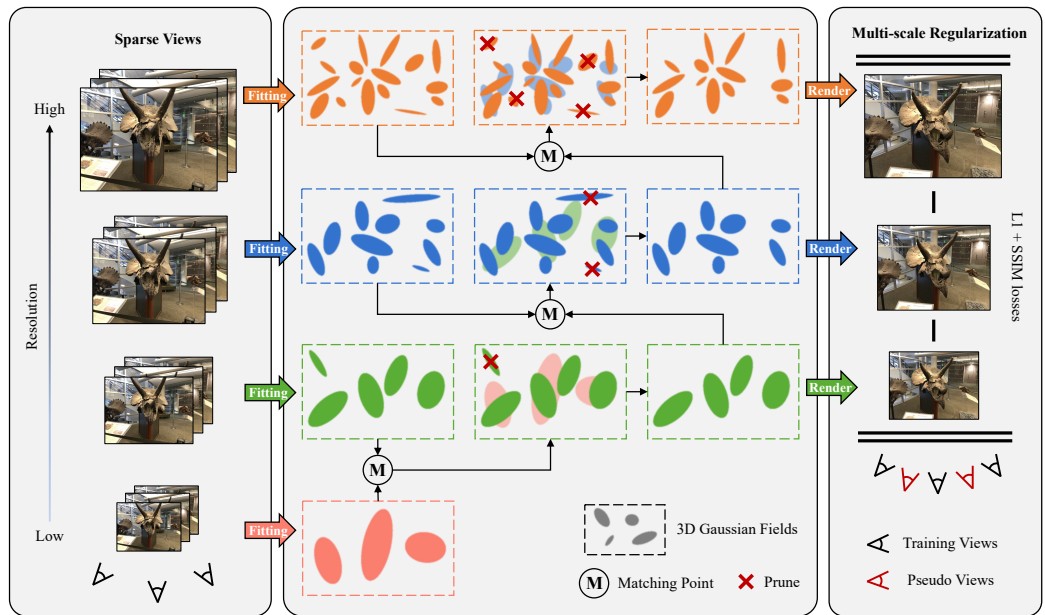

Figure 2: **Overview of our hierarchical multi-scale paradigm pipeline.** We generate multiple 3D Gaussian fields at different resolutions (left) to capture both stable global geometry and fine-grained local details. A progressive matching-based pruning strategy (middle) anchors each finer-scale field to the coarser-scale representation, filtering out noisy Gaussians while retaining essential structural details. Finally, a multi-scale pseudo-view regularization step (right) enforces cross-resolution consistency, enabling high-fidelity reconstructions.

local details from the finer resolutions while filtering out noises. This hierarchical setup establishes the foundation for the matching-based pruning in section 4.2, ensuring that every resolution contributes positively without exacerbating the pitfalls of sparse-view high-resolution reconstruction.

## 4.2 MATCHING POINT AND PRUNING

At each resolution level, the camera intrinsics are proportionally scaled, ensuring that the resulting 3D Gaussian fields remain spatially aligned within a consistent world coordinate system for further operations. To enhance robustness under sparse-view conditions, we employ a structured pruning mechanism inspired by Zhang et al. (2024), wherein poorly constrained Gaussians are systematically removed based on spatial correspondence. Specifically, we first find the matching correspondence $f : \Theta^{\text{fine}} \rightarrow \Theta^{\text{coarse}}$ for each Gaussian in the finer-scale representation $\Theta^{\text{fine}} = \{\theta_i^{\text{fine}} \mid i = 1, 2, \ldots, N^{\text{fine}}\}$ matching to its nearest counterpart in the coarser-scale field $\Theta^{\text{coarse}} = \{\theta_i^{\text{coarse}} \mid i = 1, 2, \ldots, N^{\text{coarse}}\}$:

$$f(\theta_i^{\text{fine}}) = \text{KNN}\big(\theta_i^{\text{fine}}, \Theta^{\text{coarse}}\big). \tag{1}$$

A permissible maximum distance threshold, $\tau$, is then defined, such that any finer-scale Gaussian is considered non-matching if:

$$(\theta_i^{\text{fine},x} - f(\theta_i^{\text{fine}})^x)^2 + (\theta_i^{\text{fine},y} - f(\theta_i^{\text{fine}})^y)^2 + (\theta_i^{\text{fine},z} - f(\theta_i^{\text{fine}})^z)^2 > \tau^2, \tag{2}$$

where $\theta_i^x, \theta_i^y, \theta_i^z$ denote the 3-axis positions of Gaussians. Different from CoR-GS Zhang et al. (2024) that co-prunes Gaussians within identical-scale fields using a larger threshold, we set $\tau = 3$ to ensure that filtered finer-scale Gaussians remain closely aligned with their coarser-scale anchors. This hierarchical matching and pruning strategy effectively mitigates potential overfitting and reduces noisy Gaussians before they propagate to subsequent optimization steps, while preserving consistent finer details.

Table 1: **Comparison of integrated methods.** Adding our hierarchical multi-scale paradigm to existing 3DGS-based methods consistently boosts performance, particularly at higher resolutions.

| | original Res | | | 1/2 Res | | |
|---|---|---|---|---|---|---|
| | PSNR↑ | SSIM↑ | LPIPS↓ | PSNR↑ | SSIM↑ | LPIPS↓ |
| DepthRegGS | 16.83 | 0.529 | 0.397 | 17.35 | 0.560 | 0.289 |
| HiRes + DepthRegGS | **18.52** | **0.643** | **0.233** | **19.34** | **0.687** | **0.225** |
| FSGS | 17.32 | 0.547 | 0.235 | 18.23 | 0.581 | 0.212 |
| HiRes + FSGS | **18.67** | **0.598** | **0.197** | **19.72** | **0.692** | **0.188** |
| CoR-GS | 17.51 | 0.553 | 0.287 | 18.17 | 0.575 | 0.237 |
| HiRes + CoR-GS | **19.25** | **0.686** | **0.218** | **20.03** | **0.687** | **0.194** |

## 4.3 MULTI-SCALE REGULARIZATION

We propose a multi-scale regularization process that simultaneously aligns and regularizes different resolution levels to enforce multi-scale consistency, thereby refining high-frequency details while preserving the global stability. As shown in the right section of fig. 2, we first sample the pseudo views from the two nearest training views in Euclidean space, calculating the averaged camera orientation and interpolating a virtual one between them, following the previous method Zhu et al. (2024):

$$P' = (t + \epsilon, q), \qquad (3)$$

where $t \in P$ denotes the camera location, $\epsilon \sim \mathcal{N}(0, \delta)$ is a random noise, and $q$ is a quaternion representing the interpolated rotation. These pseudo cameras are then used to render images across multiple scales, reducing the risk of overfitting.

Given that our method aims to generate high-resolution images, we enforce multi-scale consistency by ensuring that the high-resolution output remains aligned across different resolutions. Specifically, we take the highest-resolution output as the reference and downsample it to each scale $s \in \mathcal{S}$ for supervision. This ensures that the generated high-resolution output inherits structural coherence from lower-resolution scales while preserving fine details.

For each pseudo-view image $I_s^p$ rendered at scale $s \in \mathcal{S}$, we downsample the highest-resolution output $I_h^p$ to match the corresponding scale, denoted as $I_h^{p'}$. The multi-scale regularization loss is then computed as L1 reconstruction loss and D-SSIM term with the balance weight $\lambda = 0.8$:

$$R_{\text{color}}^p = \sum_{s \in \mathcal{S}} \left[ \lambda \mathcal{L}_1(I_s^p, I_h^{p'}) + (1 - \lambda) \mathcal{L}_{\text{D-SSIM}}(I_s^p, I_h^{p'}) \right]. \qquad (4)$$

At the training views, we supervise the rendered images $I_s$ with each ground-truth $I_s^*$ (downsampled from the high-resolution ground-truth $I^*$):

$$L_{\text{color}} = \sum_{s \in \mathcal{S}} \left[ \lambda \mathcal{L}_1(I_s, I_s^*) + (1 - \lambda) \mathcal{L}_{\text{D-SSIM}}(I_s, I_s^*) \right]. \qquad (5)$$

The final training loss is formulated as a weighted combination of the supervised loss from training views and the multi-scale regularization loss from pseudo-views:

$$\mathcal{L} = \mathcal{L}_{\text{color}} + R_{\text{color}}^p. \qquad (6)$$

## 5 EXPERIMENTS

### 5.1 SETTINGS

**Datasets.** We conducted experiments on the forward-facing LLFF Mildenhall et al. (2019) dataset and Mip-NeRF360 Barron et al. (2022) dataset, both constrained to a limited range of shooting angles. Our experimental setup follows prior works Niemeyer et al. (2022); Wang et al. (2023); Yang et al. (2023); Zhu et al. (2024), adopting the same data split: For LLFF, we used three views

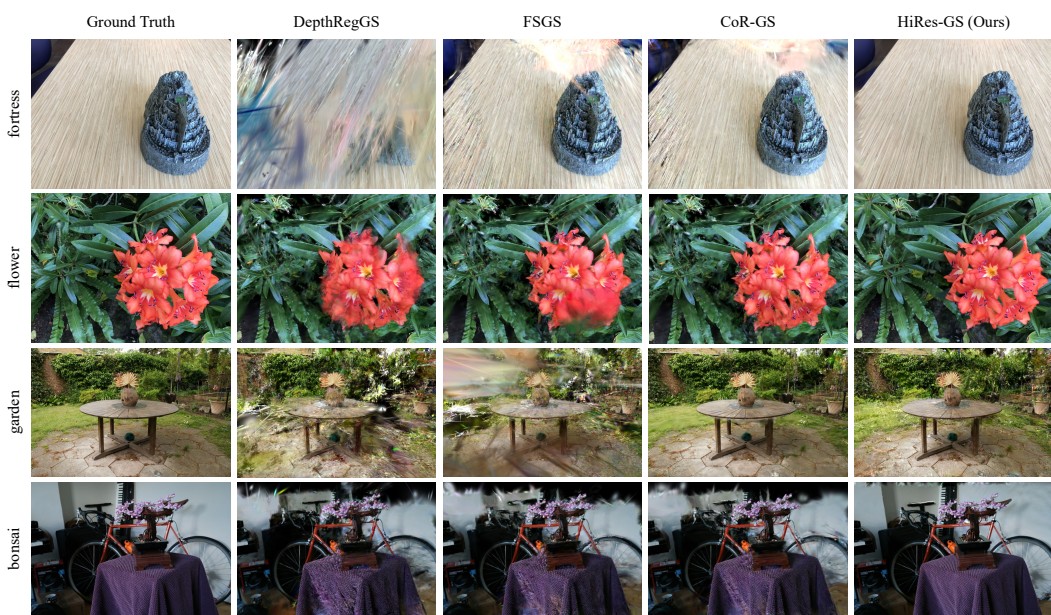

Figure 3: **Qualitative Comparison on LLFF and Mip-NeRF360 datasets.** We compare our method with current state-of-the- art few-shot reconstruction methods DepthRegGS Chung et al. (2024), FSGS Zhu et al. (2024), and CoR-GS Zhang et al. (2024). HiRes-GS yields sharper details and more stable geometry, particularly in challenging high-resolution regions.

Table 2: **Quantitative results on LLFF and Mip-NeRF360 datasets.** The best, second-best, and third-best entries are marked in red, orange, and yellow, respectively. HiRes-GS consistently achieves superior or highly competitive performance, especially at higher resolutions.

| Dataset | Resolution | Metrics | 3DGS | DepthRegGS | FSGS | CoR-GS | HiRes-GS (Ours) |
|---|---|---|---|---|---|---|---|
| LLFF | original Res | PSNR↑ | 15.38 | 15.45 | 15.48 | 15.83 | 18.45 |
| | | SSIM↑ | 0.503 | 0.537 | 0.528 | 0.553 | 0.589 |
| | | LPIPS↓ | 0.421 | 0.395 | 0.384 | 0.357 | 0.342 |
| | 1/2 Res | PSNR↑ | 16.77 | 17.20 | 17.25 | 17.51 | 19.32 |
| | | SSIM↑ | 0.527 | 0.569 | 0.577 | 0.617 | 0.633 |
| | | LPIPS↓ | 0.379 | 0.378 | 0.382 | 0.326 | 0.301 |
| | 1/4 Res | PSNR↑ | 18.67 | 18.89 | 19.54 | 19.32 | 20.18 |
| | | SSIM↑ | 0.553 | 0.584 | 0.608 | 0.645 | 0.665 |
| | | LPIPS↓ | 0.261 | 0.312 | 0.285 | 0.255 | 0.248 |
| | 1/8 Res | PSNR↑ | 19.32 | 20.13 | 20.48 | 20.45 | 20.46 |
| | | SSIM↑ | 0.649 | 0.643 | 0.687 | 0.708 | 0.703 |
| | | LPIPS↓ | 0.255 | 0.273 | 0.248 | 0.203 | 0.217 |
| Mip-NeRF360 | original Res | PSNR↑ | 15.15 | 16.07 | 15.65 | 16.85 | 18.03 |
| | | SSIM↑ | 0.457 | 0.463 | 0.438 | 0.481 | 0.527 |
| | | LPIPS↓ | 0.493 | 0.501 | 0.482 | 0.468 | 0.422 |
| | 1/2 Res | PSNR↑ | 16.11 | 16.83 | 16.26 | 17.68 | 18.74 |
| | | SSIM↑ | 0.467 | 0.473 | 0.487 | 0.510 | 0.545 |
| | | LPIPS↓ | 0.474 | 0.471 | 0.480 | 0.450 | 0.413 |
| | 1/4 Res | PSNR↑ | 17.99 | 18.19 | 18.43 | 18.68 | 19.32 |
| | | SSIM↑ | 0.511 | 0.530 | 0.502 | 0.525 | 0.557 |
| | | LPIPS↓ | 0.428 | 0.453 | 0.481 | 0.417 | 0.400 |
| | 1/8 Res | PSNR↑ | 18.80 | 18.89 | 19.57 | 19.83 | 19.81 |
| | | SSIM↑ | 0.528 | 0.556 | 0.530 | 0.567 | 0.562 |
| | | LPIPS↓ | 0.411 | 0.387 | 0.412 | 0.386 | 0.390 |

for few-shot reconstruction, and for Mip-NeRF360, we utilized 12 views. A key distinction in our approach is that we do not apply traditional downsampling to the dataset.

**Comparison Baselines.** We compare our method with the original 3DGS Kerbl et al. (2023), as well as the current state-of-the-art few-shot reconstruction methods including FSGS Zhu et al. (2024), DepthRegGS Chung et al. (2024), and CoR-GS Zhang et al. (2024).

**Evaluation Metrics.** We utilized PSNR, SSIM Wang et al. (2004), and LPIPS Zhang et al. (2018) as quantitative metrics to evaluate the reconstruction performance.

## 5.2 EXPERIMENTAL RESULTS

As shown in Table 2, our method consistently outperforms 3DGS Kerbl et al. (2023) and other state-of-the-art few-shot methods (FSGS Zhu et al. (2024), DepthRegGS Chung et al. (2024), and CoR-GS Zhang et al. (2024)) across all metrics. The improvements are particularly evident at high resolutions, where our hierarchical multi-scale framework effectively mitigates local overfitting and noise.

The qualitative results, shown in Figure 3, further demonstrate the advantages of our approach. Across all test scenes, our method consistently delivers sharper edges and more stable textures while significantly reducing artifacts such as floaters and ghosting. The effectiveness of our hierarchical pruning strategy is particularly noticeable in challenging high-resolution regions, where it successfully refines Gaussian distributions while preserving geometric coherence. Compared to other methods, FSGS Zhu et al. (2024) is susceptible to ghosting artifacts, DepthRegGS Chung et al. (2024) struggles with reconstructing stable structures due to insufficient depth constraints in high-resolution sparse-view settings, and CoR-GS Zhang et al. (2024) occasionally introduces distortions in highly detailed regions. In contrast, our hierarchical framework balances global shape consistency and local detail fidelity, resulting in more visually coherent high-resolution reconstructions.

Moreover, our hierarchical strategy can be easily integrated into existing 3DGS-based pipelines. As shown in Table 1, incorporating it into FSGS Zhu et al. (2024) and CoR-GS Zhang et al. (2024) significantly improves their performance in high-resolution settings, demonstrating the effectiveness of the hierarchical paradigm in guiding multi-scale reconstruction. These results highlight the broad applicability of our approach, offering a practical paradigm for sparse-view, high-resolution 3D Gaussian reconstruction.

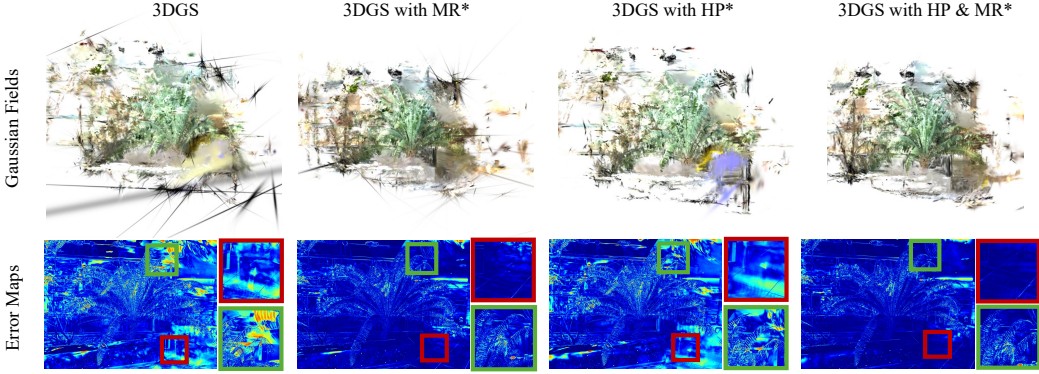

*\*MR denotes Multi-scale Regularization, HP denotes Hierarchical Pruning*

Figure 4: **Visualization results for ablation study.** Hierarchical pruning removes noisy Gaussians that deviate from the stable global structure provided by low-resolution inputs, while multi-scale pseudo-view regularization corrects local discrepancies in high-resolution reconstructions. These complementary modules synergistically enhance the overall reconstruction fidelity under sparse-view, high-resolution conditions.

## 5.3 ABLATION STUDIES

**Hierarchical Pruning and Multi-scale Regularization (table 3 and fig. 4).** We first examine the effect of our two core modules: *hierarchical pruning* and *multi-scale regularization*. From the Table 3, we can see that disabling either component leads to a noticeable drop in PSNR/SSIM, particularly at the original resolution. Combining both modules yields the best performance, confirming that pruning mitigates overfitting in the densification stage, while multi-scale regularization further refines fine-grained details without compromising global geometry. fig. 4 presents the ablated point cloud reconstructions, revealing that omitting *hierarchical pruning* leads to cluttered, noisy

Gaussians, while removing *multi-scale regularization* diminishes structure stability. These visual results corroborate the quantitative findings, illustrating that both modules are essential for robust, high-resolution reconstructions under sparse-view conditions.

**Distance Threshold in Pruning (table 4).** Next, we vary the maximum distance threshold $\tau$ used for matching finer-scale Gaussians to their coarser-scale counterparts. A stricter threshold ($\tau$=3) removes more outlier Gaussians, improving reconstruction fidelity but risking the loss of potentially useful high-frequency primitives. A looser threshold ($\tau$=10) preserves more Gaussians yet can introduce noise and floating artifacts. Empirically, a moderate setting ($\tau$=3) balances detail retention and noise suppression, yielding consistently high metrics.

**Hierarchical Recursion Depth (table 5).** We also study the effect of increasing the depth of the hierarchical recursion. As the table shows, moving from two to three scales noticeably boosts performance, indicating that additional resolution levels help capture more subtle geometric cues. However, further adding a fourth or fifth scale offers diminishing returns and increases computational overhead.

Table 3: **Ablation results of HiRes-GS.**

| Hierarchical Pruning | Multiscale Regularization | original Res | | | 1/2 Res | | |
|---|---|---|---|---|---|---|---|
| | | PSNR↑ | SSIM↑ | LPIPS↓ | PSNR↑ | SSIM↑ | LPIPS↓ |
| × | × | 15.32 | 0.512 | 0.483 | 16.28 | 0.534 | 0.395 |
| ✓ | × | 18.28 | 0.613 | 0.365 | 19.04 | 0.637 | 0.341 |
| × | ✓ | 16.43 | 0.544 | 0.408 | 16.87 | 0.565 | 0.388 |
| ✓ | ✓ | **18.79** | **0.618** | **0.337** | **19.35** | **0.646** | **0.337** |

Table 4: **Ablation results of the distance threshold.**

| Distance Threshold | original Res | | | 1/2 Res | | |
|---|---|---|---|---|---|---|
| | PSNR↑ | SSIM↑ | LPIPS↓ | PSNR↑ | SSIM↑ | LPIPS↓ |
| 1 | 18.87 | 0.603 | 0.347 | 19.30 | 0.642 | 0.280 |
| 3 | **18.89** | **0.608** | 0.332 | **19.35** | **0.643** | **0.275** |
| 5 | 18.82 | 0.598 | **0.330** | 19.32 | 0.640 | 0.277 |
| 10 | 18.63 | 0.591 | 0.376 | 19.25 | 0.627 | 0.296 |

Table 5: **Ablation results of the hierarchical recursion depth.**

| Hierarchical Recursion Depth | original Res | | | 1/2 Res | | |
|---|---|---|---|---|---|---|
| | PSNR↑ | SSIM↑ | LPIPS↓ | PSNR↑ | SSIM↑ | LPIPS↓ |
| 2 | 17.35 | 0.523 | 0.387 | 17.98 | 0.512 | 0.350 |
| 3 | 18.87 | 0.580 | 0.343 | 19.32 | 0.588 | 0.337 |
| 4 | 19.23 | **0.613** | **0.315** | **19.87** | **0.636** | **0.299** |
| 5 | **19.25** | 0.611 | **0.315** | 19.81 | 0.627 | 0.301 |

## 6 CONCLUSION

In this paper, we propose a novel hierarchical multi-scale paradigm for 3D Gaussian Splatting dubbed HiRes-GS, designed to enable high-resolution scene reconstruction under sparse-view conditions. A core innovation of our proposed method stems from our key finding that low-resolution reconstructions yield robust global geometry, whereas high-resolution inputs capture fine-grained details at the cost of increased noise and overfitting. Motivated by these complementary characteristics, our method leverages a hierarchical framework that progressively refines high-resolution details using coarser fields as stable anchors, incorporating both a matching-based pruning mechanism and a multi-scale pseudo-view regularization strategy. Experiments on the LLFF and Mip-NeRF360 datasets achieve state-of-the-art performance in high-resolution, sparse-view reconstruction, validating the effectiveness of our proposed HiRes-GS. Additionally, our paradigm can be integrated with existing 3DGS-based methods, thus extending their reconstruction capabilities from low- to all-resolutions and pushing the boundaries of current sparse-view techniques. We believe that HiRes-GS opens new avenues for high-resolution 3D scene synthesis in real-world applications.

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
