# OpenReview forum: "HiRes-GS: Hierarchical Resolution Scaling for Sparse-View High-Resolution 3D Gaussian Splatting"
_ICLR.cc/2026/Conference — ICLR 2026 Conference Withdrawn Submission_

### Official Review · Reviewer_tnJ3 · 2025-10-26

**Soundness:** 2
**Presentation:** 2
**Contribution:** 1
**Rating:** 2
**Confidence:** 5

**Summary:**

This paper presents an engineering solution for high-resolution sparse-view NVS by modifying and employing the CoR-GS framework to optimize multi-resolution fields from coarse to fine. It first gives a brief empirical observation to show the native poor quality from directly reconstructing the scene at high resolution with sparse views, and then modifies CoR-GS by adding a hierarchical structure for multi-resolution cases. Experiments are conducted on two datasets.

**Strengths:**

- The paper is easy to follow.

- A better quality in the high-resolution cases are brought.

**Weaknesses:**

- This work is a good engineering attempt on CoR-GS [1] to extend its own paradigm on the multi-resolution case, using CoR-GS's idea of cor-regularize multiple fields for a better quality in sparse-view 3DGS problem. However, it can barely be considered a sufficiently significant methodological contribution, as no new insights are brought. The target problem the authors try to express is that the high-resolution cases with more details are harder to reconstruct and synthesize, which is widely known to the community of 3D vision or even modern computer vision, and is the general inspiration for plenty of existing coarse-to-fine methods. Except for this, no more in-depth analysis or insights were provided.

- Technically, few original contributions are provided, and the authors did not clearly state the close relationship between their work to CoR-GS. Nearly all described techniques in 4.2 and 4.3 are modified from CoR-GS with the same rationale and designs (Eq (1, 2) in this paper to Eq (2, 3) in CoR-GS, and Eq (3, 4, 5, 6) to Eq (4, 5, 6, 7)). However, the authors did not clearly reveal these, just using some moderate words like "inspired" or directly ignoring the actual source when mentioning them.

- Evaluation is poor. Only two datasets with limited view selections are evaluated. The generalizability of the method for different scenes can not be proved. Also, the method is only compared to three older methods over one or two years ago (two released at the end of 2023, and one of ECCV 2024). It fails to include today's strong baselines like [2, 3, 4, 5]. The effectiveness can not be well verified.

- Results are insufficient. Only limited rendering views are provided for qualitative evaluation. Lacking common and strong evidence of the rendered depth maps or video samples, it is not possible to evaluate whether the scene geometry is correct. This raises concerns about cherry-picking. The training efficiency is also not reported.

- Transparency and fairness problems. All the implementation details for both this work itself and the reproduction of the baselines are lacking. This also raises a problem about whether the evaluations are made fairly. Especially,  considering high-resolution reconstruction is obviously different in engineering than the officially provided scripts of CoR-GS and FSGS for the low resolution case, it's not clear whether the improvements are from the multi-resolution design or just adjusting hyperparameters, such as the max iterations or some other hyperparameter changes. For example, considering much more levels of resolution are introduced in the optimization, with additional fields for each, this work may unfairly benefit from additional training iterations and longer training time with more densification. All the details should be clearly revealed.

[1] Zhang, Jiawei, et al. "Cor-gs: sparse-view 3d gaussian splatting via co-regularization." European Conference on Computer Vision. Cham: Springer Nature Switzerland, 2024.

[2] Han, Liang, et al. "Binocular-guided 3d gaussian splatting with view consistency for sparse view synthesis." Advances in Neural Information Processing Systems 37 (2024): 68595-68621.

[3] Zheng, Yulong, et al. "NexusGS: Sparse View Synthesis with Epipolar Depth Priors in 3D Gaussian Splatting." Proceedings of the Computer Vision and Pattern Recognition Conference. 2025.

[4] Park, Hyunwoo, Gun Ryu, and Wonjun Kim. "Dropgaussian: Structural regularization for sparse-view gaussian splatting." Proceedings of the Computer Vision and Pattern Recognition Conference. 2025.

**Questions:**

See the weaknesses. All the listed problems are critical.

---

### Official Review · Reviewer_4Ycz · 2025-10-31

**Soundness:** 3
**Presentation:** 3
**Contribution:** 3
**Rating:** 4
**Confidence:** 4

**Summary:**

This paper studies sparse-view 3D Gaussian Splatting (3DGS) at high output resolutions and makes the empirical observation that input resolution creates a complementary trade-off: low-resolution inputs stabilize global structure but lose high-frequency detail, whereas high-resolution inputs recover fine details but are prone to noise, ghosting, and overfitting. Building on this, the authors propose HiRes-GS, a hierarchical multi-scale paradigm that reconstructs multiple Gaussian fields at progressively higher resolutions and couples them through (1) a matching-based hierarchical pruning mechanism anchoring each finer-scale field to its coarser counterpart, and (2) a multi-scale pseudo-view regularization that renders interpolated viewpoints across scales and enforces L1 + D-SSIM consistency against the downsampled high-resolution reference. Experiments on LLFF and Mip-NeRF360 show consistent gains over 3DGS and recent sparse-view 3DGS methods; notably, the hierarchical paradigm also improves existing pipelines when plugged into FSGS, DepthRegGS, and CoR-GS. Ablations indicate both hierarchical pruning and multi-scale regularization are necessary.

**Strengths:**

1.Good writing and organization.

2.Simple and effective idea with clear motivation.

3.Plug-and-play paradigm.

**Weaknesses:**

1. Lack of comparison with other highly related works about the core novelty. The core contribution, leveraging multi-resolution Gaussian fields with coarse-to-fine guidance, resembles prior multi-level/multi-scale designs explored in the 3DGS literature, e.g., HiSplat[1] and Octree-GS[2]. Although HiSplat adopts a different feed-forward 3DGS pipeline, its core idea that low-resolution representations capture the global scene structure while high-resolution representations reconstruct fine details is largely consistent with that of this paper. Moreover, it can also be adapted to non–non-feed-forward Gaussian reconstruction pipelines, which may dilute the novelty and contribution of the proposed method. More discussions or comparisons are required to emphasize the contribution of this paper.

2. Efficiency reporting is insufficient. The method instantiates multiple Gaussian fields and introduces extra optimization (matching, pruning, pseudo-views). The paper does not report wall-clock training time, memory footprint, FLOPs, or peak Gaussian counts over training vs. baselines at each resolution, which is important for high-res practical use.

3. Missing supplement. This paper lacks supplementary material; many implementation specifics are not clear (e.g., pseudo-view sampling schedule, full pseudo code, KNN neighborhood size, initialization of Gaussians, optimizer/scheduler details). More qualitative comparisons across diverse scenes and viewpoints would also improve credibility.

[1] HiSplat: Hierarchical 3D Gaussian Splatting for Generalizable Sparse-View Reconstruction.

[2] Octree-GS: Towards Consistent Real-time Rendering with LOD-Structured 3D Gaussians.

**Questions:**

Beyond sparse input: Does HiRes-GS still provide measurable gains when the number of input images is not sparse?

---

### Official Review · Reviewer_am5k · 2025-10-31

**Soundness:** 2
**Presentation:** 2
**Contribution:** 2
**Rating:** 4
**Confidence:** 4

**Summary:**

This paper addresses sparse-view reconstruction using high-resolution images. It introduces a coarse-to-fine hierarchical Gaussian learning paradigm inspired by the observation that low-resolution Gaussians capture coarse geometry, whereas high-resolution Gaussians encode fine details. Additionally, a multi-scale pseudo-view regularization is proposed to further reduce noise and prevent overfitting.

**Strengths:**

- This paper provides clear intuition: low-resolution Gaussians capture coarse geometry, while high-resolution Gaussians encode fine details.
- Writing is generally easy to follow. The introduction is intuitive and informative (though the method section could be clearer).
- The framework can be generally applied to optimize Gaussians using sparse-view high-resolution images.

**Weaknesses:**

- While the paper is generally clear and easy to follow, it would be helpful to include a complete algorithmic overview of the entire reconstruction pipeline, as the proposed approach involves multiple optimization stages. It was somewhat difficult to grasp how the multiple Gaussian sets are optimized at a glance.
- The proposed method appears to introduce additional memory and computational overhead compared to other approaches. Since it requires optimizing multiple Gaussian sets and subsequently performing joint optimization, the overall training time is likely to be long. Furthermore, the matching-based pruning used during joint optimization may also increase computational cost, as neighbor searching is typically expensive. These factors could limit the method’s practicality in real-world applications. It would be better to include the training time in Table 2.
- The matching-based pruning removes redundant or degraded Gaussians based solely on the distance between the centers of low-resolution (e.g., 1/8 scale) and high-resolution (e.g., 1/4 scale) Gaussians. Although this criterion appears effective in practice, as shown in the ablation study, it may inadvertently discard meaningful Gaussians that are unique to the high-resolution set, either introduced during optimization or through cloning operation. Conversely, degraded Gaussians (e.g., needle-like ones) located close to low-resolution Gaussians might not be pruned.
- From my understanding and based on the paper’s findings, it is important to jointly optimize multi-scale Gaussians without allowing any single scale to dominate the optimization, as each scale serves a distinct purpose. However, the proposed multi-scale regularization uses only high-resolution renderings as pseudo targets. Since the authors note that high-resolution Gaussians do not always preserve overall geometry well, this approach might cause the low-resolution Gaussians to learn inaccurate geometry.

**Questions:**

- Why do the baseline methods perform better in the low-resolution setting (Table 2)?
- What does “hierarchical recursion depth” refer to in the ablation study? I assume it represents the number of Gaussian scales (or levels) optimized. Could the authors clarify this?
- Why does adding a fourth or fifth scale degrade performance? An intuitive explanation would be helpful for readers.

---

### Official Review · Reviewer_39m3 · 2025-11-01

**Soundness:** 3
**Presentation:** 3
**Contribution:** 3
**Rating:** 6
**Confidence:** 5

**Summary:**

This paper focuses on improving 3D Gaussian Splatting (3DGS) optimization and novel view synthesis under sparse-view settings, particularly when high-resolution images are used as input. The proposed approach generates multi-scale images by downsampling the input high-resolution images with different scaling factors for joint training. Furthermore, it introduces multi-scale point-matching pruning and pseudo-view constraints. The method is evaluated on commonly used datasets with high-resolution image inputs.

**Strengths:**

1. The proposed hierarchical pruning strategy is interesting and well-motivated, making good use of multi-scale image information. Its effectiveness is also well demonstrated through ablation studies.
2. In quantitative comparisons, the proposed method shows clear advantages over prior approaches when dealing with high-resolution image inputs.
3. The paper is well written and easy to follow.

**Weaknesses:**

1. The paper focuses on high-resolution inputs and adopts a hierarchical multi-resolution training strategy. However, similar multi-resolution training schemes have also been explored in Mip-Splatting. As both methods aim to improve robustness across different resolutions, a direct comparison with such strategies would be necessary. It also raises the question of whether combining existing methods such as FSGS or CoR-GS with the multi-resolution training in Mip-Splatting could already handle high-resolution inputs effectively.
2. The core argument in the abstract appears somewhat obvious — that low-resolution inputs lack fine details while high-resolution inputs contain more details but are harder to fit, leading to larger gradients and denser (potentially noisier) 3DGS structures.
3. The proposed multi-scale pruning is meaningful and contributes technically, but the multi-scale pseudo-view strategy seems somewhat incremental compared to prior pseudo-view constraints.

**Questions:**

I would like the authors to clarify how the proposed multi-scale pruning better than prior multi-scale joint training strategies, and to further explain the advantages of the proposed pseudo-view regularization compared to previous pseudo-view regularization methods used in fsgs and cor-gs.

---

### Note · Authors · 2025-11-12

I have read and agree with the venue's withdrawal policy on behalf of myself and my co-authors.